# Prevalence, risk factors, and treatment methods of thirst in critically ill patients: A systematic review and meta-analysis

Takuto Fukunaga[1], Akira Ouchi[2]*, Gen Aikawa[3], Saiko Okamoto[4], Shogo Uno[4], Hideaki Sakuramoto[5]

1 Department of Emergency and Critical Care Medicine, Toho University Omori Medical Center, Ota-ku, Tokyo, Japan, 2 Department of Adult Health Nursing, College of Nursing, Ibaraki Christian University, Hitachi, Ibaraki, Japan, 3 College of Nursing, Kanto Gakuin University, Yokohama, Kanagawa, Japan, 4 Department of Emergency and Critical Care Medicine, Hitachi General Hospital, Hitachi, Ibaraki, Japan, 5 Department of Critical Care and Disaster Nursing, Japanese Red Cross Kyushu International College of Nursing, Munakata, Fukuoka, Japan

* akira1Q85@gmail.com

## Abstract

Critically ill patients admitted to the intensive care unit (ICU) experience various symptoms and discomfort. Although thirst is a typical distressing symptom and should be assessed daily, it is crucial to understand its prevalence and risk factors in the ICU setting. Nevertheless, currently, systematic reviews of prevalence and risk factors are lacking. This study evaluated the prevalence and risk factors of thirst in critically ill patients. We conducted a comprehensive search of the MEDLINE, Cochrane Library, and CINAHL databases. The study design included cohort, cross-sectional, and intervention studies, including randomized and non-randomized controlled trials with control groups. The point estimates from each study were combined using a random-effects meta-analysis model. We aggregated the prevalence of thirst in ICU patients and calculated the point estimates and 95% confidence intervals. The risk of bias was assessed using the Cochrane Risk of Bias 2 tool and Newcastle-Ottawa Scale. Fifteen studies were eligible for inclusion, of which seven reported the prevalence of thirst. A total of 2,204 patients were combined, with a prevalence estimate of 0.70. The risk factors for thirst were categorized as patient and treatment factors: four patient factors (e.g., serum sodium concentration and severity of illness) and six treatment factors (e.g., nil per os and use of diuretics) were identified. However, the results showed high heterogeneity in the prevalence of thirst among critically ill patients. It was established that 70% of critically ill patients experienced thirst. Additional investigations are required to obtain a more comprehensive overview of thirst among these patients.

## Systematic review registration number

The protocol was registered in PROSPERO (ID: CRD42023428619) on June 6, 2023. (URL: https://www.crd.york.ac.uk)

**Data availability statement:** All relevant data are within the manuscript and its Supporting Information files.

**Funding:** Initials of the authors: AO Grant number: 24K20319 The full name of funder: JSPS KAKENHI URL of funder website: https://www.jsps.go.jp The funding source had no role in the design, practice,

**Competing interests:** The authors have declared that no competing interests exist.

## Introduction

Critically ill patients in the intensive care unit (ICU) experience various symptoms and discomfort [1,2], and recently, thirst has been highlighted as a stressful symptom. In their study, Chanques et al. [3] reported that thirst is one of five ICU patient symptoms that should be evaluated daily because it is the most prevalent and intense symptom reported by ICU patients. This statement is supported by various studies that highlight that thirst is related to significant distress and stress, increased oxygen consumption and metabolic load on organs [4,5], and induced delirium [6], which all affect recovery.

The prevalence of thirst in critically ill patients varies between studies. Some studies [3,4] reported a prevalence of thirst in severely ill patients ranging from 40–70%. However, there are no meta-analyses addressing thirst in critically ill patients and therefore the prevalence is unknown.

In addition, risk factors and treatment methods for thirst in critically ill patients remain unclear. Thirst is associated with xerostomia, endotracheal tubes, tracheostomies, oxygen therapy, solid and liquid fasting, electrolyte alterations, and hypovolemia. In a study conducted in three ICUs in a tertiary medical center [7], oral swab wipes, sterile ice water spray, and lip balm significantly reduced thirst intensity and distress. In a survey of 61 ICU patients [8], vitamin C sprays, peppermint water mouthwash, and a lip moisturizer significantly reduced thirst intensity. Both studies recommended combining these measures as an oral care bundle. However, few studies have examined interventions for thirst, and risk factors and effective treatments are unknown.

Therefore, we conducted a systematic review and meta-analysis of thirst in critically ill patients to examine the prevalence and risk factors thereof. The secondary aim was to identify assessment and treatment methods. Our findings will contribute to a better clinical understanding of thirst in critically ill patients.

## Materials and methods

### Protocol and registration

This systematic review and meta-analysis examined the prevalence of thirst, assessment methods, risk factors, and symptom relief methods in critically ill patients. The study methodology followed the Preferred Reporting Items for Systematic Reviews and Meta-Analyses Statement [9], and a systematic review was conducted. The protocol was registered in PROSPERO (ID: CRD42023428619) on June 6, 2023 (S1 File). The systematic review adhered to the PRISMA 2020 checklist (S2 File).

### Literature search strategy

We searched for articles in three electronic databases: MEDLINE via PubMed, the Cochrane Library (CENTRAL), and CINAHL. We also conducted a manual search using Google Scholar. All English publications published until June 6, 2024 were searched without any restrictions on countries. The reference lists of all selected articles were independently screened to identify additional studies that were excluded from the initial search. The search formulae for PubMed, CENTRAL, and CINAHL are presented in Supplementary S3 File.

### Study screening and selection

Two of the five reviewers (TF, GA, SO, SU, and HS) independently screened the titles and abstracts of all studies to identify potentially relevant studies. Subsequently, the full texts were independently reviewed according to a standardized protocol. Any disagreements between the

two reviewers were resolved through discussion, and if necessary, a third reviewer (AO) was brought in for arbitration.

## Eligibility criteria

To conduct a comprehensive and exploratory review of thirst in critically ill patients, we used the "population–concept–context" framework [10] recommended by the Joanna Briggs Institute as a guide for developing clear and meaningful objectives and eligibility criteria; The study population included critically ill adult patients expected to stay in the ICU for more than 24 hours, the concept was any treatment for thirst or dry mouth, and the context was critically ill patients in ICUs. The study design included cohort, cross-sectional, and prospective studies, including randomized controlled trials (RCTs) and non-RCTs (NRCTs) with control groups. Only studies reporting the prevalence of thirst, risk factors, and treatment methods for thirst in critically ill patients, respectively; that did not include preclinical results; were in English; and that included human participants were included.

## Data extraction and quality assessment

The prevalence of thirst was recorded as the primary outcome of each study. Secondary evaluations were collected from the literature and included assessment methods and risk factors for thirst, and symptom relief methods. Data extraction was performed by TF. Data were extracted using Microsoft Excel. Each article was summarized under the following data headings in an Excel spreadsheet: First author; Year of publication; Title of Article; Study characteristics; Sample size; Control group; Prevalence; Demographic factors (including average age, severity of illness, etc.); Symptom relief methods; Risk factors; Assessment methods; and Definition of thirst.

The risk of bias in the target literature was assessed using the Cochrane Risk of Bias 2 tool (RoB2) for RCTs [11], and the study design's quality and degree of potential bias were assessed according to the RoB2 domains. Two reviewers (TF and AO) independently evaluated the risk of bias using the RoB2. Disagreements between the two reviewers were discussed and if unresolved, a third reviewer (HS) was brought in for arbitration. The Newcastle-Ottawa Scale was used for observational studies, cross-sectional studies, and NRCTs. For RCTs, we assessed each of the following six domains: (1) selection bias in the process of randomization assignment, (2) selection bias in the process of concealment of assignment, (3) implementation bias in the process of blinding to participants and interventionists, (4) measurement bias in the process of blinding to outcome assessors and analysts, (5) missing bias due to participant attrition, and (6) reporting bias in the reporting of outcomes were assessed. For observational studies, cross-sectional studies, and NRCTs, we assessed each of the following three domains: (1) representativeness, exposure assessment, etc. (four items), (2) confounding assessment (two items), and (3) outcomes (three items). For observational studies, cross-sectional studies, and NRCTs, the risk of bias was assessed according to each of the following three domains: (1) representativeness, exposure assessment, etc. (four items), (2) confounding assessment (two items), and (3) outcomes (three items). The risk of bias for the Newcastle-Ottawa Scale was assessed by summing the scores of the nine criteria to evaluate the overall quality of each study; if the two raters had different opinions, a decision was made through an inter-rater discussion.

## Statistical analysis

Analyses were conducted via Stata/BE (version 18.0; Stata Corp., College Station, TX, USA). The point estimates from each study were combined using a random-effects meta-analysis

model to obtain an overall estimate using the Der Simonian–Laird method. The Freeman–Tukey double-arcsine transformation was chosen as an effect size measure. The estimate was considered significant when the Z test p-value was < .05. We assessed heterogeneity with Cochrane's Q test and tau-squared ($T^2$) and measured the inconsistency (the percentage of total variation across studies due to heterogeneity) of effects across interventions using the $I^2$ statistic. Publication bias was assessed using a funnel plot and Egger's test, with significance set at P < .05. In addition, the prevalence of each definition of thirst sensation was extracted by subgroup analysis. A meta-regression analysis to determine the contribution of specific cofactors (mean age, sex ratio, mechanical ventilator use, and length of ICU stay) to heterogeneity was conducted.

## Results

### Literature search results

The literature search results are summarized in Fig 1. A total of 1,267 references were retrieved from the three English databases. A total of 1,093 references were targeted for primary screening after excluding duplicates. Subsequently, 100 studies were selected and evaluated for full text as secondary screening. Fifteen eligible studies met the inclusion criteria.

### Selected literature characteristics

After the screening, 15 studies were selected for eligibility. Regarding the study design, nine studies were observational [4,12–19], two studies were cross-sectional [6,20], and four studies were RCTs [7,8,21,22]. Six studies were from the United States [4,7,12,13,16,20], two each from France [14,22], Japan [6,15], and China [8,18], and one each from Italy [17], South Korea [21], and Norway [19]. Two references were published in the 2000s [12,13], seven in the 2010s [4,6,7,14,16,20,21], and six in the 2020s [8,15,17–19,22].

Regarding patient characteristics of the selected references, five studies included only patients on ventilators [13,16,17,19,21] and four studies had a mean patient age of 65 years or older [6,12,14,15]. Furthermore, eight references were mixed ICUs [6,7,13,15,18–20,22], two were internal medicine ICUs [12,21], three were general ICUs [8,14,17], one was a surgical ICU [16], and another was an ICU with no type description [4]. Some internal medicine ICUs were included in the literature, thus limiting their scope to patients with cancer (Table 1).

### Risk of bias assessment

Risk of bias was assessed using the RoB2 and the Newcastle-Ottawa Scale and are summarized in Table 2. Finally, two studies were judged to be high risk, ten to be with some concerns, and three to be low risk.

Two reviewers independently assessed the risk of bias for each included study. When the Cochrane Collaboration Risk of Bias Tool was used, the overall risk of bias was determined by following the tool's standardized instructions: studies assessed to have "low" risk of bias in all tool domains were considered to have "low" overall risk, studies assessed to have "some" risk of bias in at least one tool domain were considered to have "some" overall risk, and studies assessed to have "high" risk of bias in at least one domain *or* assessed to have "some" risk of bias in two or more domains in a way that "substantially lowers confidence in the result" were considered to have "high" overall risk. When the Newcastle-Ottawa Scale was used, the overall risk of bias was determined by the average score of the two reviewers: greater than or equal to seven was considered as having "low" overall risk, greater than or equal to four but less than seven was considered as having "some" overall risk, and less than four was considered having "high" overall risk.

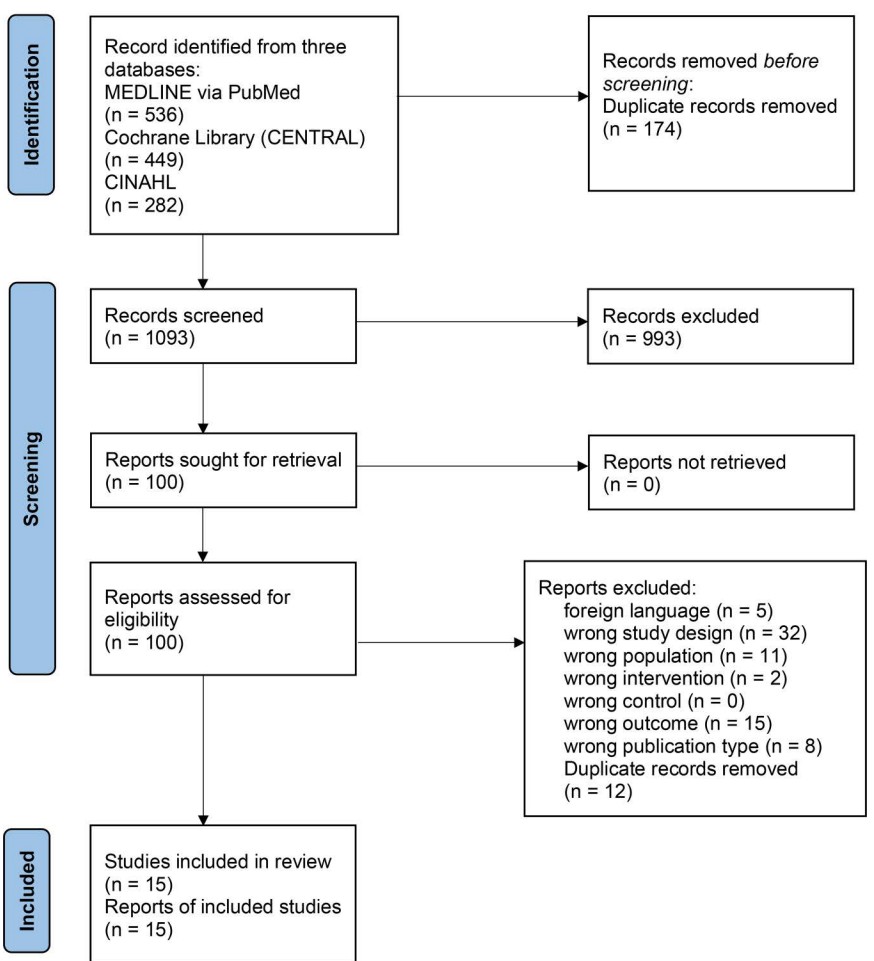

**Fig 1. Preferred reporting items for systematic reviews and meta-analyses statement flowchart.**

## Primary outcomes

The prevalence of thirst is summarized in Fig 2. Excluded studies in this analysis are summarized in S1 and S2 Tables. Eight studies [4,6,12,15,17–20] reported the prevalence of thirst drawing on data from 2,204 patients, resulting in a prevalence estimate of 0.70 (95% CI: 0.59–0.79), with high statistical heterogeneity ($T^2 = 0.09$, $I^2 = 95.45\%$, P <.001). When stratified by the definition of thirst, there were four references with a numerical rating scale (NRS) score ≥ 1 or with other scales to determine the presence of thirst. The prevalence estimate was 0.71 (95% CI: 0.67–0.76), with low statistical heterogeneity ($T^2 = 0.00$, $I^2 = 37.96\%$, P =.18). Three references with an NRS score ≥ 3 had a prevalence estimate of 0.77 (95% CI: 0.66–0.86), with high statistical heterogeneity ($T^2 = 0.03$, $I^2 = 88.45\%$, P <.001). One had an NRS score ≥ 8 and a prevalence estimate of 0.41 (95% CI: 0.36–0.46); heterogeneity could not be assessed. For excluded study is summarized in S4 File.

The risk factors for thirst are summarized in Table 3. Risk factors for thirst were divided into patient and treatment factors. Four patient factors were identified: serum sodium

**Table 1. Study characteristics.**

| Number | Author_year | Country | Study design | Purpose | Inclusion criteria | Key findings |
|---|---|---|---|---|---|---|
| 1 | Nelson JE_2001 [12] | USA | Prospective observational study | To clarify the experience of symptoms of ICU patients at high risk of in-hospital mortality | Cancer patients admitted to MICU | Using the Edmonton Symptom Assessment Scale, ICU patients at high risk of in-hospital mortality experience pain, discomfort, anxiety, sleep disturbances, and hunger and thirst |
| 2 | Li DTY_2007 [13] | USA | Prospective, descriptive study | To determine cardiovascular and pupillary reactivity and behavioral responses during distressing and non-distressing procedures in sedated ICU patients | • Patients aged between 21 and 80 years<br>• Mechanical ventilated for more than 12 hours<br>• Understand English<br>• Ramsay score of 2 | Specific physiological responses and changes in pupil size are potentially useful nociceptive indicators in the ICU setting |
| 3 | Puntillo KA_2010 [4] | USA | Prospective, observational study | To assess the symptom experience of ICU patients at high risk of death and evaluate the relationship between delirium and patient symptoms | • Patients 18 years or older admitted to the ICU for at least 3 days<br>• APACHE II score of ≧ 20 in the first 24 hours<br>• One or more of the following diagnoses: acute heart failure, respiratory failure, chronic liver failure with cirrhosis, multiorgan failure, sepsis, or systemic failure associated with a diagnosis of malignancy | • Patients at high risk of death experienced pain, tiredness, shortness of breath, restlessness, anxiousness, sadness, hunger, scared, thirst and confusion with a prevalence of thirst of 70.8<br>• Patients with delirium were significantly more likely to report confusion (43% vs. 22%, p = .004) and sadness (46% vs. 31%, p = .04) than those without delirium |
| 4 | Siami S_2013 [14] | France | Prospective interventional study | To examine how vasopressin secretion during an osmotic challenge (500 mL of hypertonic saline administered for 120 minutes) is a variable response in patients recovering from septic shock | Adult patients hospitalized with septic shock who survived five days after the discontinuation of vasopressor therapy | 60% of patients recovering from septic shock had no response to vasopressin secretion during osmotic challenge |
| 5 | Doi S_2021 [15] | Japan | Cohort study | To investigate the impact of oral care on the assessment of thirst and dry mouth in ICU patients | Patients ≧ 18 years of age | The NRS for thirst decreased from immediately to 1 hour after oral care |
| 6 | Duffy EI_2018 [16] | USA | Pilot prospective study | Validate devices that aid communication, such as communication boards and eye tracking devices | • Mechanical ventilated patients<br>• Speaks English<br>• No cognitive impairment<br>• No traumatic brain injury with motor impairment | • The median accuracy of the communication board was (100% [IQR 100%-100%]) and the eye tracking device had a median accuracy of (100% [IQR 68.8%-100%]) |
| 7 | Jang CS_2016 [21] | South Korea | Randomized clinical trial | To evaluate the combined effect of oral healthcare on oral health status, dry mouth, and salivary pH in critically ill and intubated patients | Intubated patients over 20 years of age | • Beck Oral Health Assessment Scale: (8.1 vs. 11.1, P <.001)<br>• Dry mouth (sublingual gland): (2.0 vs. 1.0, P <.001)<br>• Dry mouth (sublingual gland):(1.0 vs. 0.0, P <.001)<br>• Saliva pH: (6.5 vs. 5.7, P <.001) |
| 8 | Puntillo KA_2014 [7] | USA | Single-blinded, randomized clinical trial | To examine the effects of intervention bundles on thirst intensity, thirst distress, and dry mouth | • Patients aged over 18 years<br>• ICU stay for more than 24 hours<br>• Speaks English<br>• Can give name, date and location<br>• RASS of -1 to + 1<br>• Either thirst intensity or thirst distress score at screening is NRS 3 or higher | Comparing the mean NRS estimates for the control and intervention groups, the intensity of thirst was 3.6 vs. 4.7 and the distress of thirst was 3.2 vs. 3.7 |

*(Continued)*

**Table 1.** (Continued)

| Number | Author_year | Country | Study design | Purpose | Inclusion criteria | Key findings |
|---|---|---|---|---|---|---|
| 9 | Stotts NA_2015 [20] | USA | Descriptive cross-sectional study | To identify predictors of presence, intensity, and distress of dry mouth in ICU patients | • Patients over 18 years of age<br>• ICU stay for more than 24 hours<br>• Speaks English<br>• Can give name, date, and location<br>• RASS of -1 to +1<br>NRS of thirst is more than 3 | • Predictors for the presence of thirst were APACHE II (P = .030), NPO (P = .006), high-dose opioid use (P = .001), high-dose furosemide use (P = .038), selective serotonin reuptake inhibitor use (P = .048) and ionized calcium (P = .008)<br>• Predictors for thirst intensity were oral intake (P = .001), gastrointestinal disease (P = .020), and NPO (P = .001)<br>• Predictors for thirst distress were mechanical ventilator (P = .014), fluid balance (P = .043), antihypertensive medication (P = .001), gastrointestinal disease (P = .003), and NPO (P = .002) |
| 10 | Sato K_2019 [6] | Japan | Single-center retrospective cross-sectional study | To determine if persistent severe thirst is strongly associated with the onset of delirium | Patients ≧ 18 years of age<br>RASS of -1 to +1 | • The prevalence of strong thirst with NRS 8 or higher was 40.6%, and strong thirst lasting more than 24 hours was associated with delirium (odds ratio, 5.74; 95% confidence interval, 2.53-12.99) |
| 11 | Negro A_2022 [17] | Italy | Prospective observational study | To investigate the incidence of thirst sensation in ICU patients and assess the association between dry mouth sensation and endotracheal tube, tracheostomy, spontaneous breathing, and oxygen therapy with or without humidification | Patients 18 years of age or older with tracheal intubation or tracheostomy and spontaneous respiration<br>GCS greater than or equal to 9 | The incidence of thirst in ICU patients is as high as 76.1% and is associated with high-dose diuretics, NPO, and dry mouth |
| 12 | Merliot-Gailhoustet L_2022 [22] | France | Cross-over randomized controlled trial | To examine the best ways to improve ICU patient discomfort among various electronic relaxation devices | Patients ≧ 18 years of age<br>CAM-ICU negative with RASS greater than or equal to 0<br>SOFA score of 3 or higher | Virtual reality devices were associated with a significant reduction in overall discomfort (median NRS = 4[2–6] vs. 2[0-5]; p =.01, mixed-effect model), indicating that virtual reality devices, compared to music therapy, were most effective in improving overall discomfort and reducing physiological stress responses in ICU patients. and physiological stress reactions in ICU patients. |
| 13 | Zhang W_2022 [8] | China | Prospective, randomized, placebo-controlled | To demonstrate the effectiveness of an intervention bundle to reduce thirst and dry mouth | • 18 years of age or older<br>• ICU stay for more than 24 hours<br>• Fasting patients<br>• Clear conscious and able to cooperate<br>• NRS of thirst measured by screening is greater than or equal to 3 | Mean decrease in thirst intensity and objective oral mucosa scale after intervention decreased more for patients in the intervention group compared to controls (-1.27 vs. -0.19, -0.36 vs. -0.1) |
| 14 | Lin R_2023 [18] | China | Prospective descriptive design | To analyze the incidence of and factors contributing to dry mouth in ICU patients with and without dry mouth by analyzing differences in physiological, psychological, disease- and environment-related parameters | • Patients 18 years of age or older<br>• ICU stay for more than 24 hours<br>• RASS of -1 to +1<br>• Able to communicate verbally and understand the questionnaire<br>• Consent to participate in the study | The incidence of thirst in ICU patients is as high as 76.1% and is associated with high-dose diuretics, NPO, and dry mouth |

*(Continued)*

**Table 1.** (Continued)

| Number | Author_year | Country | Study design | Purpose | Inclusion criteria | Key findings |
|---|---|---|---|---|---|---|
| 15 | Saltnes-Lillegård C_2024 [19] | Norway | Prospective cohort study | To describe the prevalence, intensity and distress of five symptoms in ICU patients and to investigate possible predictive factors associated with symptom intensity | • Patients 18 years of age or older<br>• ICU stay for more than 24 hours<br>• Need for mechanical ventilation<br>• Need for continuous vasoactive therapy or ICU stay greater than 24 hours | • On the first ICU day, thirst as the most prevalent symptom (66%), with the highest mean intensity score<br>• On the seven ICU days, thirst as the most prevalent symptom (64%), with the highest mean intensity score<br>• During seven days, analgesic administration and sepsis diagnosis were associated with increased thirst intensity |

*Note:* ICU, intensive care unit; MICU, medical intensive care unit; RASS, Richmond agitation-sedation scale; APACHE, acute physiologic assessment and chronic health evaluation; SOFA, sequential organ failure assessment; NRS, numerical rating scale; NPO, nil per os.

**Table 2. Quality rating of studies reporting.**

| Author_year | Bias tool used | Average Newcastle-Ottawa Scale score (if applicable) | Overall risk of bias |
|---|---|---|---|
| Nelson JE_2001 | Newcastle-Ottawa Scale | 4 | Some concerns |
| Li DTY_2007 | Newcastle-Ottawa Scale | 5 | Some concerns |
| Puntillo KA_2010 | Newcastle-Ottawa Scale | 6 | Some concerns |
| Siami S_2013 | Newcastle-Ottawa Scale | 7 | Low risk |
| Doi S_2021 | Newcastle-Ottawa Scale | 7 | Low risk |
| Duffy EI_2018 | Newcastle-Ottawa Scale | 5 | Some concerns |
| Jang CS_2016 | Cochrane Risk of Bias 2 | | Some concerns |
| Puntillo KA_2014 | Cochrane Risk of Bias 2 | | Some concerns |
| Stotts NA_2015 | Newcastle-Ottawa Scale | 5 | Some concerns |
| Sato K_2019 | Newcastle-Ottawa Scale | 5 | Some concerns |
| Negro A_2022 | Newcastle-Ottawa Scale | 3 | High risk |
| Merliot-Gailhoustet L_2022 | Cochrane Risk of Bias 2 tool for cross-over trials | | High risk |
| Zhang W_2022 | Cochrane Risk of Bias 2 | | Low risk |
| Lin R_2023 | Newcastle-Ottawa Scale | 5 | Some concerns |
| Saltnes-Lillegård C_2024 | Newcastle-Ottawa Scale | 6 | Some concerns |

concentration, hyperglycemia, illness severity, and xerostomia. Six treatment factors were identified: nil per os, surgery, diuretics, humidified Venturi mask, serotonin reuptake inhibitors, and high doses of opioids in morphine equivalents (>50 mg).

## Secondary outcomes

The methods used to evaluate thirst are summarized in Table 4. Thirteen studies reported methods of assessing thirst, with NRS being the most frequently evaluated method by nine references [6,7,8,13,15,17,18,23]. The visual analog scale (VAS) was reported by one study [14], the Edmonton Symptom Assessment Scale by two studies [4,12], the Vidatak EZ picture board by one study [16], Patient Symptom Survey symptom checklist [19] and the Tobii Dynavox I-15 eye tracking device by one study [16].

Symptom relief methods for thirst are summarized in Table 5. Five articles reported methods for alleviating the symptoms of thirst. Doi et al. [15] examined changes in the NRS for

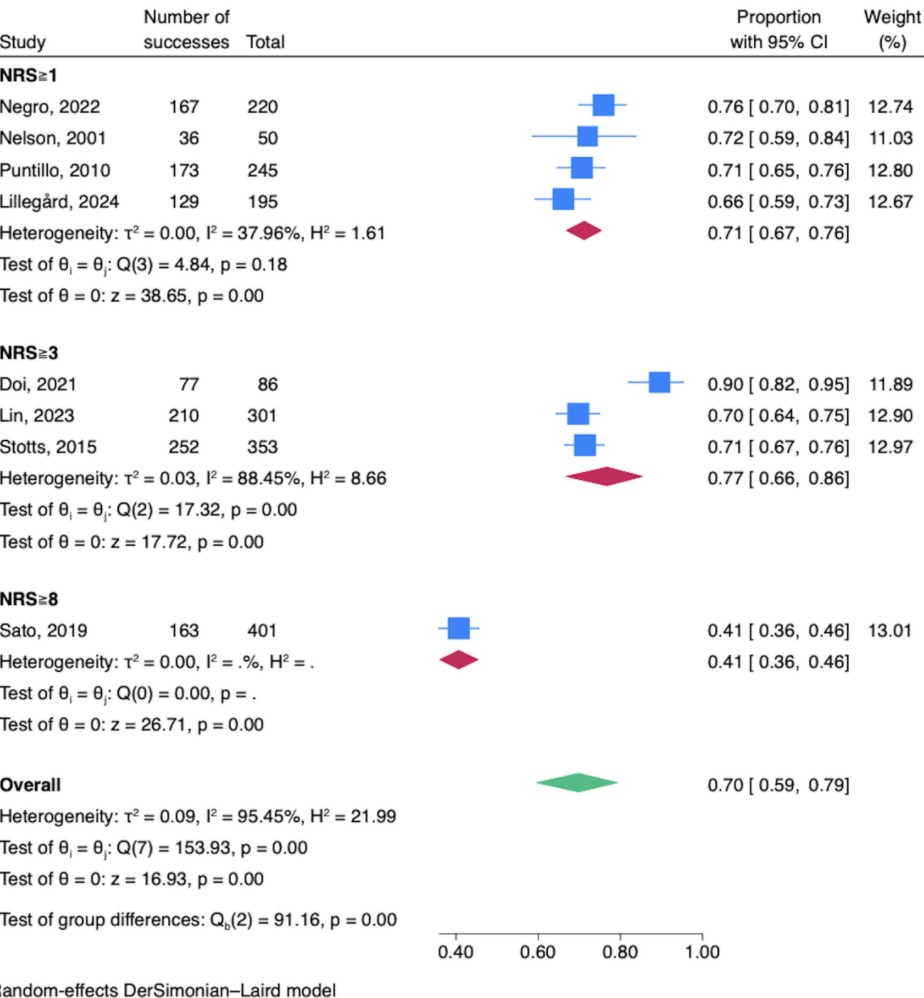

**Fig 2. Forest plot for prevalence of thirst sensation.**

**Table 3. Risk factors for thirst in critically ill patients.**

| Patient factors | Treatment factors |
| --- | --- |
| Serum sodium concentration | Nil per os |
| Hyperglycemia | Surgery |
| Illness severity | Diuretic |
| Xerostomia | Humidified venturi mask |
| | Serotonin reuptake inhibitors |
| | High doses of opioids in morphine equivalents (> 50 mg) |

thirst, lip moistness, and the Modified Revised Oral Assessment Guide after providing oral care every four hours to the intervention group. The results showed a significant effect on the NRS of thirst immediately after oral care (-1 [-3 to 0], $P < .01$) to 1 h later (0 [-1.25 to 1], $P = 0.04$). Jang et al. [21] performed tooth brushing, swabbing with 0.1% chlorhexidine, and intermittent swabbing with cold water for the intervention group and conventional oral care

**Table 4. Assessment method and definition of thirst sensation included in the systematic review.**

| Measurement tool | Scale | Thirst definition | Number of studies |
|---|---|---|---|
| NRS | 0–10 point scale | None, NRS ≥ 1, NRS ≥ 3, NRS ≥ 8 | 9 |
| Visual analog scale | 0–100 or 0–10 point scale | None | 1 |
| Edmonton Symptom Assessment Scale | Yes/no response | Yes | 2 |
| Patient Symptom Survey symptom checklist | Yes/no response | Yes | 1 |
| Vidatak EZ picture board | Yes/no response | Yes | 1 |
| The Tobii Dynavox I-15 eye tracking device | Follow a cursor with their eyes | | 1 |

*Note*: NRS, numerical rating scale

**Table 5. Symptom relief methods of thirst in critically ill patients.**

| Author_year | Sample size | Intervention group | Control group | Intervention | Comparator |
|---|---|---|---|---|---|
| Doi S_2021 | 86 | | | Oral care every 4 hours | None |
| Jang CS_2016 | 35 | 18 | 17 | Twice a day at 05:00 and 17:00. Oral care Swabbing with 0.1% chlorhexidine Intermittent swabbing with cold water | The control group received conventional oral care with gauze and saline twice daily (at 05:00 and 17:00) |
| Merliot-Gail-houstet L_2022 | 60 | HEALTHY-MIND© VR system: 53 DEEPSEN© VR system:56 MUSIC-CARE: 55 | Normal Care: 54 | Virtual reality devices and music therapy | TV and radio |
| Puntillo KA_2014 | 252 | 127 | 125 | Apply menthol moisturizer to patients' lips using an oral swab and water spray for 15 minutes | 15 minutes of normal care |
| Zhang W_2022 | 61 | 31 | 30 | Intervention was performed from 8:00 to 18:00 for three days Moisturized oral cavity with vitamin C spray (concentration: 10 mg/mL) every hour, mouth rinse with peppermint water (5 g of peppermint + 50 mL of slightly warm water cooled to 40°C) at 14:00, and apply moisturizer (main ingredient: glycerin) to lips | Intervention was performed from 8:00 to18:00 for three days. Saline spray every hour to moisturize the oral cavity, mouthwash (50 mL of hot water cooled to 40°C), moisturizing the mouth with slightly warm water every 2 hours, and routine oral care consisted of wiping with a saline cotton ball and rinsing with warm water |

using gauze and saline for the target group, and compared them with the Beck Oral Health Assessment Scale. The results showed significant effects on oral status (8.1 vs. 11.1, P <.001), dry mouth (sublingual: 2.0, 1.0, P <.001; dorsal tongue: 1.0, 0.0, P <.001), and pH (6.5, 5.7, P <.001). Puntillo et al. [7] compared the intensity of thirst and distress in an intervention group with 15 min of oral swabbing, water spray, and application of menthol moisturizer to patients' lips and in a control group with 15 min of usual care. The results showed significant effects on both thirst intensity (-2.3 vs. -0.6) and distress (-1.8 vs. -0.4). In a RCT [8], the intervention group received hourly oral moisturization with vitamin C spray, mouthwash with peppermint water, and application of lip balm. Oral moisturizer with saline spray every hour, mouthwash with warm water, and lip moisturizer with warm water every 2 h were given to the control group, and NRS of thirst feeling and the objective oral mucosa scale were compared. The study results showed significant effects on both the NRS of thirst (-1.27 vs. -0.19, P <.001) and the objective oral mucosa scale (-0.36 vs. 0.1, P = .008). In addition, there were four references that used oral care bundles and interventions. On the other hand, Merliot-Gailhoustet et al.

[22], who assessed pain, anxiety, thirst, dyspnea, and insomnia with virtual reality equipment and music therapy for the intervention group and usual care, such as television and radio for the target group, did not find a significant reduction in thirst.

### Meta-regression analysis

Meta-regression analyses indicated no significant difference in the prevalence of thirst based on mean age, sex ratio, and mechanical ventilator use (S3 Table). We found a negative association between the overall prevalence of thirst and ICU length of stay.

### Publication bias

The funnel plot is shown in S1 Fig. However, Egger's test results did not uncover any significant indications of publication bias (z = 1.38, p =.17).

## Discussion

A systematic review and meta-analysis of critically ill patients admitted to the ICU synthesized the prevalence of thirst in critically ill patients quantitatively from 14 references and qualitatively synthesized it regarding assessment, risk factors, and symptom relief methods. There were seven references on prevalence, indicating a very high prevalence rate of 70%. The most common evaluation method was the NRS, and the cut-off values for thirst differed in different studies. Furthermore, some studies classified and evaluated thirst sensation based on distress and intensity. There were seven references on risk factors, categorized into patient and treatment factors. Among the items listed as risk factors, serum sodium level, severity of illness, nil per os, diuretic administration, and tracheal intubation were the most common and highly relevant for critically ill patients. There were five references on symptom relief methods, all showing significant results related to oral care. Some references examined interventions to promote oral moistening and salivation using oral care bundles.

### Prevalence of thirst in critically ill patients

The prevalence of thirst in critically ill patients was very high in this study. To our knowledge, this is the first systematic review to focus on the prevalence of thirst in critically ill patients. Konstam et al. [23] reported a 16% prevalence of thirst in medically treated patients with stable heart failure. Lee et al. [24] also reported a prevalence of moderate to severe thirst of 55.8% in postoperative patients. Compared to these studies, the prevalence of thirst in critically ill patients was higher.

Mechanisms can be considered regarding risk factors for thirst sensation in critically ill patients. The reasons for thirst in critically ill patients include an environment in which they cannot drink freely, such as during fasting, as well as a deficiency of intracellular fluid due to elevated plasma osmolality, extracellular fluid due to decreased blood volume, and mouth dryness [25–28]. An increase in osmolality is believed to cause the release of antidiuretic hormones from the posterior pituitary gland by stimulation of osmotic receptors in the anteroventral wall of the third ventricle of the hypothalamus, which sends stimuli to the anterior cingulate cortex and cerebellum, causing a feeling of thirst. ICU patients are prone to hyperglycemia [29], hypernatremia has been reported in approximately 26% of ICU patients [30], and diuretics are associated with increased osmolality. As for the decreased blood volume, stimulation of the arterial and baroreceptors of the heart and kidneys, resulting in atrial natriuretic peptide inhibition and sodium and water reabsorption, may send stimuli to the anterior cingulate cortex and cerebellum, causing a sense of thirst. Generally, ICU patients are prone to thirst sensations due to fluid balance changes, such as bleeding, vomiting, diarrhea, sweating,

and diuresis. When the mouth is opened by oral tracheal intubation, xerostomia may cause the secretion of antidiuretic hormone from peripheral osmotic receptors in the oropharynx, which sends stimuli to the anterior cingulate cortex and cerebellum, causing a sensation of thirst. Regarding opioid effects, which are associated with the body's water regulation system and may affect thirst perception [31,32], selective serotonin reuptake inhibitor suppress electrical activity in the subfornical organ at the entrance to the third ventricle; therefore, these drugs may act on this forebrain structure to promote sodium intake and thirst [33].

Regarding humidified Venturi masks, there is the impression that heating and humidifying is a positive effect, however, these masks are a risk factor for thirst sensation. Although fasting, eating, drinking, and other factors may have an influence in the background, heating and humidifying alone is unlikely to lead to a reduction in thirst. Based on the above findings, it is likely that the difference in prevalence is due to more factors that cause thirst in severely ill patients. Therefore, it is necessary to determine the prevalence of thirst at an early stage using appropriate assessment methods.

## The assessment methods of thirst in critically ill patients

Based on the study results, the cut-off for the presence or absence of thirst using the NRS varies from NRS 1 or higher, NRS 3 or higher, to NRS 8 or higher, and the definition of thirst is not well defined. Owing to the subjective nature of the symptoms, it is difficult to define a clear cut-off. In any case, the optimal assessment of symptoms should include intensity, frequency, and distress [34], and should quantify more than just the presence or absence of thirst. Furthermore, in a systematic review examining strategies in the management of postoperative thirst [35,36], the NRS or VAS was the most common method of assessment. Similar to pain, the NRS or VAS method for evaluating thirst, a subjective symptom, is considered the gold standard. Recent studies have attempted to quantify the distress caused by dry mouth in detail using the Thirst Discomfort Scale, a 12-item five-point scale [37]. Thus, it is important to appropriately assess thirst for early detection, prevention, and symptom relief in critically ill patients.

## The symptom relief methods of thirst in critically ill patients

The mechanism of thirst is thought to stem from a deficiency of extracellular and intracellular components and from stimuli sent from the oropharyngeal region to peripheral osmotic receptors. Studies examining the mechanism of thirst have reported [27] that cold stimulation of the oropharyngeal region is more useful than correction of dehydration in improving thirst. The results indicated that methods to alleviate thirst included oral moisturizing and ice water, citric acid, and ascorbic acid for pH adjustment, use of peppermint and menthol to provide a cool sensation, and lip moisturizing. Cold water ingestion may be more effective than room temperature water in relieving thirst as a stimulus to the oropharyngeal region [38,39], and ice water is representative of direct cold stimulation of the oropharynx. Other interventions, like citric acid and ascorbic acid, can promote salivation and alleviate thirst [40]. It has been confirmed that peppermint and menthol act directly on the oral mucosa and provide long-lasting cold stimulation, which promotes saliva secretion, retaining moisture in the mucous membrane, and alleviating the feeling of thirst [28,40]. Although the intervention methods are different, this intervention can achieve the effects of both cold stimulation and saliva secretion. Regarding lip moisturization, frequent glycerin application can maintain mucosal homeostasis [41]. Therefore, these mechanisms suggest that the symptom relief methods qualitatively integrated into this study were effective for thirst.

Furthermore, symptomatic relief from thirst requires a bundle of multiple interventions rather than a single one. In their study of symptom relief, Puntillo et al. [7] compared the

intensity of thirst and distress in an intervention group with 15 min of oral swabbing, water spray, and application of menthol moisturizer to patients' lips and in a control group with 15 min of usual care. They reported a 2.3 point reduction in thirst intensity and a 1.8 point reduction in thirst distress. In Zhang et al.'s study [8], the intervention group received hourly oral moisturization with vitamin C spray, mouthwash with peppermint water, and application of lip balm. Oral moisturizer with saline spray every hour, mouthwash with warm water, and lip moisturizer with warm water every 2 h were given to the control group. They reported a 1.27 point decrease in the NRS score for dry mouth. For patients experiencing pain, a decrease of approximately two points in the NRS score is considered clinically important [42]. Therefore, Puntillo et al. may highlight an effective intervention because it reduced the intensity of thirst by more than two points. However, a comparison of the post-care NRS scores between the intervention and control groups revealed a decrease of only 1.1 points. Thus, differences in intervention bundles, intervention procedures, timing of evaluation, inclusion and exclusion criteria, and other factors make it difficult to directly compare results across studies; however, no single bundled intervention would be able to accomplish much if it were less effective.

## Limitations and implications

A strength of this study is that it is the first systematic review and meta-analysis to validate thirst in critically ill patients, allowing us to estimate its prevalence. However, the results showed high heterogeneity in the prevalence of thirst among critically ill patients. We estimated high heterogeneity based on the definition of thirst, with the group defined as NRS 3 or higher showing the greatest variability. In addition, meta-regression analyses results indicated that only length of ICU stay showed significant differences. However, further research is needed to fully explain the heterogeneity. Possible reasons for this include differences in patient populations, assessment methods, timing, inclusion and exclusion criteria, and other factors. Therefore, caution should be exercised when interpreting the integrated results regarding the prevalence of thirst in critically ill patients. As a challenge for future research, new surveys and studies on this subject are needed to obtain a more comprehensive overview as relevant studies are scarce. Second, since the study results showed high heterogeneity in the prevalence of thirst in critically ill patients, discussions about the definition of thirst, sample size, and inclusion criteria should be considered. Third, the criteria for the presence of thirst are defined differently by different studies, and it is unclear which values are clinically meaningful. Therefore, basic research is needed to unify thirst definitions in the future.

## Conclusion

From the study results, it was established that 70% of critically ill patients experienced thirst. To minimize the distress caused by thirst, it is necessary to understand the pathophysiology and risk factors, properly assess the early detection of symptoms, and implement symptom relief using oral care bundles. Moreover, further investigations are needed to obtain a more comprehensive overview of thirst among critically ill patients.

## Supporting information

**S1 File. PROSPERO protocol.**
(PDF)

**S2 File. Preferred reporting items for systematic reviews and meta-analyses statement checklist.**
(PDF)

**S3 File. Search formulas.**
(PDF)

**S4 File. Quality assessment of studies.**
(PDF)

**S1 Table. Description of inclusion/exclusion articles.**
(PDF)

**S2 Table. Data extraction of included articles in the systematic review and meta- analysis.**
(PDF)

**S3 Table. Results of the meta-regression analysis.**
(PDF)

**S1 Fig. Funnel plot.**
(PDF)

## Acknowledgments

We would like to thank Editage (www.editage.jp) for English language editing.

## Author contributions

**Conceptualization:** TAKUTO FUKUNAGA, Akira Ouchi.

**Data curation:** TAKUTO FUKUNAGA, Akira Ouchi.

**Formal analysis:** TAKUTO FUKUNAGA, Akira Ouchi.

**Funding acquisition:** Akira Ouchi.

**Investigation:** TAKUTO FUKUNAGA, Akira Ouchi, Gen Aikawa, Saiko Okamoto, Shogo Uno, Hideaki Sakuramoto.

**Methodology:** Akira Ouchi, Gen Aikawa, Hideaki Sakuramoto.

**Project administration:** TAKUTO FUKUNAGA, Akira Ouchi, Hideaki Sakuramoto.

**Supervision:** Akira Ouchi.

**Validation:** TAKUTO FUKUNAGA.

**Visualization:** Akira Ouchi.

**Writing – original draft:** TAKUTO FUKUNAGA, Akira Ouchi, Gen Aikawa, Hideaki Sakuramoto.

**Writing – review & editing:** TAKUTO FUKUNAGA, Akira Ouchi, Gen Aikawa, Saiko Okamoto, Shogo Uno, Hideaki Sakuramoto.

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

## Further readings

Agrawal V, Agarwal M, Joshi SR, Ghosh AK. Hyponatremia and hypernatremia: disorders of water balance. J Assoc Physicians India. 2008;56:956–64

Clark J, Archer SK. Thirst interventions in adult acute care-what are the recommended management options and how effective are they? A systematic review. Dimens Crit Care Nurs. 2022;41(2):91–102. https://doi.org/10.1097/DCC.0000000000000511

Devlin JW, Skrobik Y, Gélinas C, Needham DM, Slooter AJC, Pandharipande PP, et al. Clinical practice guidelines for the prevention and management of pain, agitation/sedation, delirium, immobility, and sleep disruption in adult patients in the ICU. Crit Care Med. 2018;46(9):e825–e873. https://doi.org/10.1097/CCM.0000000000003299

Higgins J, Thomas J, Chandler J, Cumpston M, Li T, Page M, et al. Chapter 10. Analysing data and undertaking meta-analyses. In: Cochrane handbook for systematic reviews of interventions. version 6.4; 2010. [cited 2024 Jan 30]. Available from: https://training.cochrane.org/handbook/current

Ho V, Goh G, Tang XR, See KC. Underrecognition and undertreatment of thirst among hospitalized patients with restricted oral feeding and drinking. Sci Rep. 2021;11(1):13636. https://doi.org/10.1038/s41598-021-93048-4

Kress JP, Hall JB. Sedation in the mechanically ventilated patient. Crit Care Med. 2006;34:2541–6. https://doi.org/10.1097/01.CCM.0000239117.39890.E3

Landström M, Rehn IM, Frisman GH. Perceptions of registered and enrolled nurses on thirst in mechanically ventilated adult patients in intensive care units-a phenomenographic study. Intensive Crit Care Nurs. 2009;25(3):133–9. https://doi.org/10.1016/j.iccn.2009.03.001

McKinley MJ, Cairns MJ, Denton DA, Egan G, Mathai ML, Uschakov A, et al. Physiological and pathophysiological influences on thirst. Physiol Behav. 2004;81:795–803. https://doi.org/10.1016/j.physbeh.2004.04.055

Rose L, Nonoyama M, Rezaie S, Fraser I. Psychological wellbeing, health related quality of life and memories of intensive care and a specialised weaning centre reported by survivors of prolonged mechanical ventilation. Intensive Crit Care Nurs. 2014;30(3):145–51. https://doi.org/10.1016/j.iccn.2013.11.002

Rosner MH, Ronco C. Dysnatremias in the intensive care unit. Contrib Nephrol. 2010;165:292–8. https://doi.org/10.1159/000313769

Schittek GA, Simonis H, Bornemann-Cimenti H. Pain, nausea, vomiting, thirst, cold, … the challenge of well-being in post-operative patients. Intensive Crit Care Nurs. 2021;66:103090. https://doi.org/10.1016/j.iccn.2021.103090

So HM, Chan DSK. Perception of stressors by patients and nurses of critical care units in Hong Kong. Int J Nurs Stud. 2004;41:77–84. https://doi.org/10.1016/S0020-7489(03)00082-8

Stachenfeld NS, DiPietro L, Nadel ER, Mack GW. Mechanism of attenuated thirst in aging: role of central volume receptors. Am J Physiol. 1997;272:R148–R157. https://doi.org/10.1152/ajpregu.1997.272.1.R148

Waldréus N, Hahn RG, Jaarsma T. Thirst in heart failure: a systematic literature review. Eur J Heart Fail. 2013;15:141–9. https://doi.org/10.1093/eurjhf/hfs174

