## [Decision Letter · Decision Letter 0]

25 Sep 2024

PONE-D-24-28619Prevalence, risk factors, and treatment methods of thirst in critically ill patients: A systematic review and meta-analysisPLOS ONE

Dear Dr. FUKUNAGA,

Thank you for submitting your manuscript to PLOS ONE. After careful consideration, we feel that it has merit but does not fully meet PLOS ONE’s publication criteria as it currently stands. Therefore, we invite you to submit a revised version of the manuscript that addresses the points raised during the review process.

We look forward to receiving your revised manuscript.

Kind regards,

Rajeev Singh

Academic Editor

PLOS ONE

Journal Requirements:

“NO authors have competing interests.”

“Initials of the authors: AO

Grant number: 24K20319

The full name of funder: JSPS KAKENHI

URL of funder website: https://www.jsps.go.jp

The funding source had no role in the design, practice,”

5. As required by our policy on Data Availability, please ensure your manuscript or supplementary information includes the following:

Reviewers' comments:

Reviewer's Responses to Questions

**Comments to the Author**

1. Is the manuscript technically sound, and do the data support the conclusions?

Reviewer #1: Yes

2. Has the statistical analysis been performed appropriately and rigorously? 

Reviewer #1: No

3. Have the authors made all data underlying the findings in their manuscript fully available?

Reviewer #1: Yes

4. Is the manuscript presented in an intelligible fashion and written in standard English?

Reviewer #1: Yes

5. Review Comments to the Author

Reviewer #1: Summary and general comments:

In the manuscript “Prevalence, risk factors, and treatment methods of thirst in critically ill patients: A systematic review and meta-analysis” the authors performed a literature search followed by meta-analyses of 15 published studies.

The main results showed:

1. An overall pooled frequency of thirst estimated at 70%

2. A high overall between-studies heterogeneity (I2 = 95%) due, at least in part, to the definition of thirst

Overall, the paper is well written, straight forward, simple, clear and concise, and I really enjoyed reading it, however it needs some rectifications.

Specific comments:

#1 Statistical analysis: The statistical analysis section is incomplete and requires:

a) Definition of the effect size (proportion or logit transformed proportion, etc.) together with the Z test and the threshold of significance (presumably 0.05).

b) Analyzing between-studies heterogeneity through I2 is insufficient. I2 tell us nothing about the dispersion of effects across included studies. The authors should mention the Q test and the value of Tau2, and calculate the 95% prediction intervals (95% PI). The authors should not confuse the 95% PI with the 95% CI. The authors can use the CMA prediction intervals free software to compute all 95% PI through the following link https://meta-analysis-workshops.com/pages/predictionintervals. The calculation of the 95% PI is simple and based on the following 4 items: effect-size, upper bound of the 95% CI, Tau2 and number of included studies.

c) Publication bias assessment through Egger’s test and the generation of a funnel plot.

#2 Sources of heterogeneity: The results showed that there was a huge amount of between-studies heterogeneity. The authors performed a subgroup analysis by “definition of thirst” based on a “numerical rating scale” which could explain at least in part the between studies heterogeneity. If possible/applicable the authors could improve their study through:

a- Subgroup analysis by ethnicity.

b- Meta-regressions using patients’ mean age, gender-ratio (M/F), condition, duration of hospital stay, and any relevant clinical or biological feature in critically ill patients.

This will enable the authors to enrich the results with more insightful information about sources of the observed heterogeneity.

#3 Publication bias: The authors did not perform a publication bias analysis. The authors should:

a- Add a sub-chapter to the “statistical analysis” section, namely “Publication bias assessment” and describe the procedure. For instance: Publication bias was assessed by Egger’s test and visualized through the generation of a funnel plot.

b- Add a sub-chapter to the “Results” section in which the authors should depict the results of publication bias for each comparison.

c- Add a figure of a Funnel plot.

Thanks for the opportunity to review this manuscript. If appropriately corrected, this meta-analysis could provide valuable information on the frequency of thirst in critically ill patients, the risk factors and patients management.

6. PLOS authors have the option to publish the peer review history of their article (what does this mean? ). If published, this will include your full peer review and any attached files.

**Do you want your identity to be public for this peer review?** For information about this choice, including consent withdrawal, please see our Privacy Policy .

Reviewer #1: **Yes: ** Tarak Dhaouadi

---

## [Author Response · Author response to Decision Letter 1]

20 Nov 2024

November 20, 2024

Dr. Rajeev Singh

Academic Editor

PLOS ONE

Dear Dr. Singh,

Thank you for inviting us to submit a revised draft of our manuscript, “Prevalence, risk factors, and treatment methods of thirst in critically ill patients: A systematic review and meta-analysis,” to PLOS ONE (Manuscript ID: PONE-D-24-28619). We appreciate the time and effort you and the reviewer dedicated to providing insightful feedback on ways to strengthen our paper. Thus, it is with great pleasure that we resubmit our manuscript for further consideration. We incorporated changes that reflect the detailed suggestions you graciously provided. To facilitate your review, our point-by-point responses are provided below.

Response to Reviewer

Comment #1—Statistical analysis: The statistical analysis section is incomplete and requires:

a) Definition of the effect size (proportion or logit transformed proportion, etc.) together with the Z test and the threshold of significance (presumably 0.05).

Response: Thank you for pointing this out. We added the following to our revised Methods section: “The Freeman–Tukey double-arcsine transformation was chosen as an effect size measure. The estimate was considered significant when the Z test p-value was < .05. We assessed heterogeneity with Cochrane’s Q test and tau-squared (T2) and measured the inconsistency (the percentage of total variation across studies due to heterogeneity) of effects across interventions using the I2 statistic.”

b) Analyzing between-studies heterogeneity through I2 is insufficient. I2 tell us nothing about the dispersion of effects across included studies. The authors should mention the Q test and the value of Tau2, and calculate the 95% prediction intervals (95% PI). The authors should not confuse thec with the 95% CI. The authors can use the CMA prediction intervals free software to compute all 95% PI through the following link https://meta-analysis-workshops.com/pages/predictionintervals. The calculation of the 95% PI is simple and based on the following 4 items: effect-size, upper bound of the 95% CI, Tau2 and number of included studies.

Response: Thank you for providing this feedback. We calculated the 95% prediction intervals, which ranged from 1.375–2.577. The Cochrane’s Q test and tau-squared (T2) have been added to the revised Results section as follows: “Eight studies [4,6,12,15,17-20] reported the prevalence of thirst drawing on data from 2,204 patients, resulting in a prevalence estimate of 0.70 (95% CI: 0.59–0.79), with high statistical heterogeneity (T2 = 0.09, I2 = 95.45%, P < .001). When stratified by the definition of thirst, there were four references with a numerical rating scale (NRS) score ≥ 1 or with other scales to determine the presence of thirst. The prevalence estimate was 0.71 (95% CI: 0.67–0.76), with low statistical heterogeneity (T2 = 0.00, I2 = 37.96%, P = .18). Three references with an NRS score ≥ 3 had a prevalence estimate of 0.77 (95% CI: 0.66–0.86), with high statistical heterogeneity (T2 = 0.03, I2 = 88.45%, P < .001).”

c) Publication bias assessment through Egger’s test and the generation of a funnel plot.

Response: Thank you for pointing this out. We added the following sentence to our revised Methods section: “Publication bias was assessed using a funnel plot and Egger’s test, with significance set at P < .05.”

Comment #2—Sources of heterogeneity: The results showed that there was a huge amount of between-studies heterogeneity. The authors performed a subgroup analysis by “definition of thirst” based on a “numerical rating scale” which could explain at least in part the between studies heterogeneity. If possible/applicable the authors could improve their study through:

a- Subgroup analysis by ethnicity.

Response: Thank you for pointing this out. Owing to a lack of data, we could not analyze ethnicity. Analyses in other areas were conducted based on data availability.

b- Meta-regressions using patients’ mean age, gender-ratio (M/F), condition, duration of hospital stay, and any relevant clinical or biological feature in critically ill patients.

This will enable the authors to enrich the results with more insightful information about sources of the observed heterogeneity.

Response: Thank you for pointing this out. A meta-regression analysis was performed and Table S6 was added. We also revised the Results section as follows: “Meta-regression analyses indicated no significant difference in the prevalence of thirst based on mean age, sex ratio, and mechanical ventilator use (S2 Table). We found a negative association between the overall prevalence of thirst and ICU length of stay.”

Comment #3—Publication bias: The authors did not perform a publication bias analysis. The authors should:

a- Add a sub-chapter to the “statistical analysis” section, namely “Publication bias assessment” and describe the procedure. For instance: Publication bias was assessed by Egger’s test and visualized through the generation of a funnel plot.

Response: Thank you for the suggestion. As noted above, we added the following to our Methods section: “Publication bias was assessed using a funnel plot and Egger’s test, with significance set at P < .05.”

b- Add a sub-chapter to the “Results” section in which the authors should depict the results of publication bias for each comparison.

Response: Thank you for pointing this out. We planned to test for publication bias when the number of articles used in the meta-analysis was 10 or more. In each subgroup comparison, we did not carry out additional analysis due to the limited number of studies.

We revised our Results section as follows: “The funnel plot is shown in S1 Fig. However, Egger’s test results did not uncover any significant indications of publication bias (z = 1.38, p = .17).”

c- Add a figure of a Funnel plot.

Thanks for the opportunity to review this manuscript. If appropriately corrected, this meta-analysis could provide valuable information on the frequency of thirst in critically ill patients, the risk factors and patients management.

Response: Thank you again. As noted above, we added a funnel plot.

We hope our manuscript is now suitable for publication in PLOS ONE.

Sincerely,

The Authors

---

## [Editor Report · Decision Letter 1]

27 Nov 2024

Prevalence, risk factors, and treatment methods of thirst in critically ill patients: A systematic review and meta-analysis

PONE-D-24-28619R1

Dear Dr. FUKUNAGA,

We’re pleased to inform you that your manuscript has been judged scientifically suitable for publication and will be formally accepted for publication once it meets all outstanding technical requirements.

Kind regards,

Rajeev Singh

Academic Editor

PLOS ONE
---

## [Editor Report · Acceptance letter]

PONE-D-24-28619R1

PLOS ONE

Dear Dr. Ouchi,

I'm pleased to inform you that your manuscript has been deemed suitable for publication in PLOS ONE. Congratulations! Your manuscript is now being handed over to our production team.

Kind regards,

on behalf of

Dr. Rajeev Singh

Academic Editor

PLOS ONE